# Synergetic Prototype Learning Network for Unbiased Scene Graph Generation

## ABSTRACT

Scene Graph Generation (SGG) is an important cross-modal task in scene understanding, aiming to detect visual relations in an image. However, due to the various appearance features, the feature distributions of different categories have suffered from a severe overlap, which makes the decision boundaries ambiguous. The current SGG methods mainly attempt to re-balance the data distribution, which is dataset-dependent and limits the generalization. To solve this problem, a Synergetic Prototype Learning Network (SPLN) is proposed here, where the generalized semantic space is modeled and the synergetic effect among different semantic subspaces is delved into. In SPLN, a Collaboration-induced Prototype Learning method is proposed to model the interaction of visual semantics and structural semantics. The conventional visual semantics is focused on with a residual-driven representation enhancement module to capture details. And the intersection of structural semantics and visual semantics is explicitly modeled as conceptual semantics, which has been ignored by existing methods. Meanwhile, to alleviate the noise of unrelated and meaningless words, an Intersection-induced Prototype Learning method is also proposed specially for conceptual semantics with an essence-driven prototype enhancement module. Moreover, a Selective Fusion Module is proposed to synergetically integrate the results of visual, structural, conceptual branches and the generalized semantics projection. Experiments on VG and GQA datasets show that our method achieves state-of-the-art performance on the unbiased metrics, and ablation studies validate the effectiveness of each component.

## CCS CONCEPTS

• **Computing methodologies → Scene understanding**.

## KEYWORDS

Scene Graph Generation, Prototype Learning, Cross-modal integration, Generalized Semantic Space.

## 1 INTRODUCTION

Scene Graph Generation (SGG) is attracting more and more researchers as a critical cross-modal task in the field of scene understanding. The scene graph is a powerful structural knowledge because it creatively converts an image into a graph that can be

*ACM MM, 2024, Melbourne, Australia*
© 2024 Copyright held by the owner/author(s). Publication rights licensed to ACM.
ACM ISBN 978-x-xxxx-xxxx-x/YY/MM
https://doi.org/10.1145/nnnnnnn.nnnnnnn

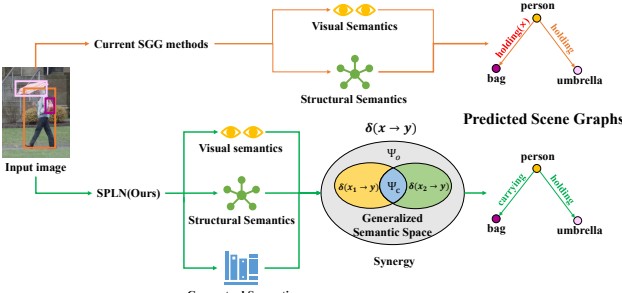

**Figure 1: The paradigm of current methods and our method. Current SGG methods simply fuse the visual and structural semantics, while our SPLN reorganizes three different semantics in the generalized semantic space and explores the synergetic effect among them comprehensively. The region marked in yellow, green, blue, and gray denotes the visual semantic subspace, structural semantic subspace, conceptual semantic subspace, and the other semantics such as linguistics and knowledge, respectively.**

easily modeled and understood by machines. Scene graphs are composed of visual triplets in the form of <subject-predicate-object>, in which the subject and the object are denoted as nodes and the predicate is denoted as edge. Many downstream tasks can benefit from scene graphs, such as image captioning [4, 10], human-object interaction detection [9], and visual question answering [8, 13].

However, the development of SGG has suffered from severe highly-skewed long-tailed bias, that is, predicates in the majority are trivial and less informative, such as "on" and "in". On the contrary, the more meaningful predicates with rich semantic information, such as "watching" and "attached to", are less in quantity. Therefore, the predicted logit exhibits a similar long-tailed distribution, which undermines the application of SGG. To address the long-tailed problem, many methods [21, 26, 45] have attempted to re-balance the data distribution in the dataset. However, these methods rely heavily on statistics prior and relieve the bias by defining various rules that involve excessive human intervention, which makes them sensitive to datasets and limits the generalization. Furthermore, the ambiguous decision boundaries caused by diverse appearance features make the long-tailed problem even worse, which is ignored by the above methods.

As Fig.1 shown, existing methods [7, 41, 46] only fuse visual semantics and structural semantics to obtain the final result, and the holistic supervision signal brings noises during training. Specifically, the most practice is to integrate the visual features of triplets with certain word meanings of entities, which is supervised by a one-hot vector of target predicate label. This makes current methods predict the correct predicate between "person" and "umbrella" as shown in the upper right corner of Fig.1. However, this holistic

supervision signal is easily affected by long-tailed bias, and confuses the method to align visual semantics and structural semantics. The training samples of "holding" is 6.5 times of "carrying", and that is why the current SGG method makes the wrong prediction "person-holding-bag" as Fig.1 shown.

To address this issue, in this paper, the generalized semantic space is reorganized to delve into the synergetic effect among different semantic subspaces, and each semantic subspace is guided by a suitable supervision signal separately. Visual semantics and structural semantics are two components involved in the generalized semantic space. There also exist other semantic subspaces such as linguistic semantics, knowledge semantics, etc. In SGG, the most commonly adopted semantics are visual and structural semantics. The visual semantics includes the scene contexts and pairwise entity interaction, etc, and the structural semantics includes word embedding of labels, knowledge graphs, etc. By analyzing these two types of semantics, we find there is an overlap between them. For example, there are various appearance features of pairwise entities corresponding to the predicate "behind", and this word also has its own word embedding in low-dimensional textual space. Although these different representations are in different semantic subspaces, they all point to the same concept, i.e. *At or to the back or far side of*. Therefore, conceptual semantics should also be modeled explicitly. Specifically, the definitions in Wiktionary are utilized to model this conceptual semantics. Compared with the predicates between "person" and "bag" predicted by the current SGG method, the ambiguity is relieved by our method. The concept of "holding" is *grasp or grip*, and the concept of "carrying" is *lift and take it to another place*, this difference is captured by our method to predict correctly.

In Fig.1, a venn diagram depicts the semantic subspaces of two sources $x = \{x_1, x_2\}$ influencing the target distribution $y$. The visual semantics is marked in yellow, and the distribution based on this is denoted as $\delta(x_1 \rightarrow y)$. Similarly, the structural semantics is marked in green and the distribution is denoted as $\delta(x_2 \rightarrow y)$. The whole synergetic effect in the generalized semantic space is decomposed as $\delta(x \rightarrow y) = \delta(x_1 \rightarrow y) + \delta(x_2 \rightarrow y) + \psi_c + \psi_o$, where $\delta(x_1 \rightarrow y) + \delta(x_2 \rightarrow y)$ denotes the interaction. $\psi_c$ denotes the intersection, which is marked in blue and represents our newly proposed conceptual semantics. $\psi_o$ is the complementary set, which is marked in gray. The universal set is generalized semantic space.

Therefore, a Synergetic Prototype Learning Network (SPLN) is proposed here, which is composed of a Collaboration-induced Prototype Learning branch (CPL), an Intersection-induced Prototype Learning branch (IPL), and a Selective Fusion Module (SFM). In CPL, visual semantics is extracted and embedded as prototypes. Due to the diversity of appearance features, more details are hoped to be preserved to increase discrimination when generating representations for classification. Thus, a Residual-Driven Representation Enhancement (RDRE) module is designed. In IPL, conceptual semantics is also modeled as prototype. To weaken the impact of irrelevant words, an Essence-Driven Prototype Enhancement (EDPE) module is designed to capture the most critical part of each concept. Finally, the SFM models the integration of distributions from different semantics according to the decomposed generalized semantic space, where a simple but effective loss on the similarity matrix of prototypes is designed to drive them discriminative.

Overall, the main contributions of this paper are:

- A Synergetic Prototype Learning Network (SPLN) is proposed to deal with the ambiguity decision boundaries based on the decomposed generalized semantic space of SGG.
- We propose a Collaboration-induced Prototype Learning method with a residual-driven representation enhancement module to preserve details of representations in the visual semantic space.
- We propose an Intersection-induced Prototype Learning method with an essence-driven prototype enhancement module to alleviate unrelated noises in the conceptual semantic space.
- A Selective Fusion Module is proposed to model the distribution based on three semantic branches and the generalized semantics projection. Moreover, our SPLN achieves state-of-the-art performance on Visual Genome and GQA datasets.

## 2 RELATED WORK

### 2.1 Unbiased Scene Graph Generation

SGG has been troubled by the long-tailed problem for a long time. Since Tang et al. [35] designed the unbiased metric mean Recall@K, the unbiased SGG was formally defined to deal with this problem. To deal with this problem, many re-sampling and re-weighting strategies are proposed, which are dependent on statistics priors. Li et al. [25] devised a bi-level data re-sampling strategy to alleviate the imbalanced data distribution in the dataset. Min et al. [30] adopted a re-sampling strategy using the median amount of the samples over all predicate classes. Biswas et al. [1] addressed the insufficient samples of minority predicate classes by borrowing samples of minority classes from its neighboring triplet in the semantic space. However, the development of unbiased SGG should not be limited to these re-balance strategies. The unique characteristics of scene graphs need to be focused on, that is, the ambiguous decision boundaries caused by diverse visual features. Our SPLN provides a promising way to solve this problem by modeling the generalized semantic space of SGG and delving into the interaction and intersection between different semantic subspaces.

### 2.2 Prototype Learning

Prototype learning is applied in various fields, such as image classification [48], recommendation systems [12], clustering analysis [29], etc. In the field of SGG, prototype learning has been explicitly introduced since Zheng et al. [47]. They devised prototype-guided learning and prototype regularization to match relation representations with corresponding predicates. After that, prototype-based methods are attracting more and more researchers. Li et al. [22] measured the invariance of each predicate class for unbiased prototypes. Li et al. [27] generated attended semantic prototype representations for each triplet sample, capturing the intrinsic visual patterns. Our SPLN is also a prototype-based method, which introduces a new conceptual prototype to represent the conceptual semantics.

## 3 METHOD

### 3.1 Problem Definition

Following previous works [7, 35, 45], the framework of our method is two-stage. The object detector is followed by SGG modules, which

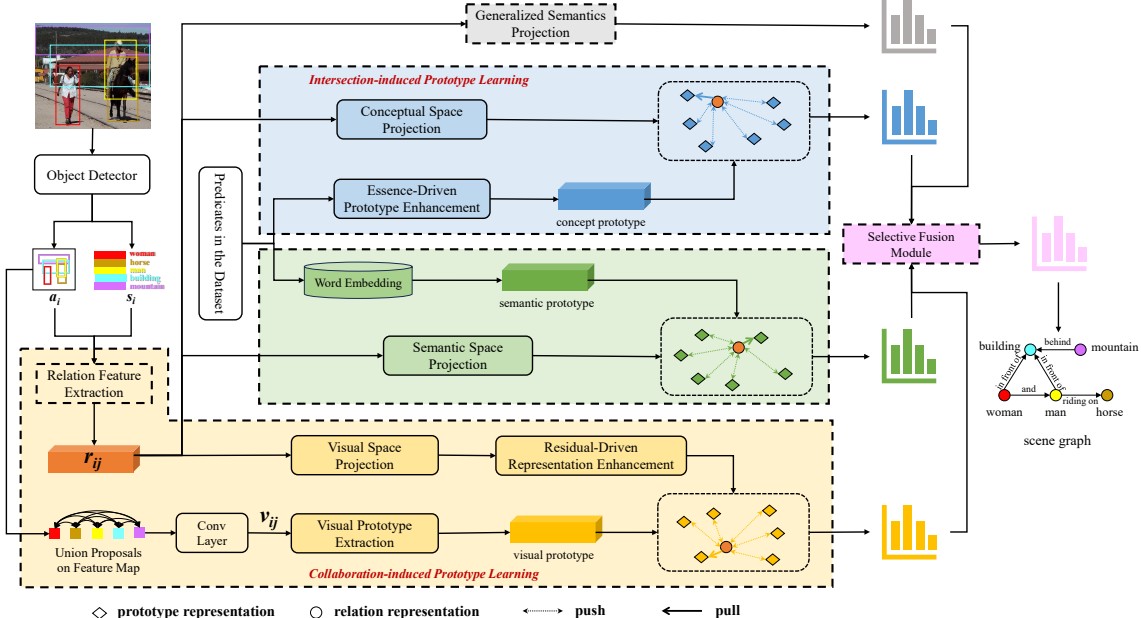

**Figure 2: The overview of our proposed SPLN, which can be roughly divided into five parts, i.e. the generalized semantics projection module, three parallel branches to model the prototype learning within the corresponding semantic subspace, and the selective fusion module to integrate the distributions from the decomposed generalized semantic space.**

can utilize the semantic information provided by entity labels. Here we choose Faster R-CNN [34] as our object detector, and the whole process of SGG is reconsidered from the new perspective, which is formulated as follows:

$$P(SG|I) = P(R|S_v) + P(R|S_t) + P(R|S_c) + P(R|S_o), \quad (1)$$

where $SG$ denotes the generated scene graph, $I$ denotes the input image, and $R$ denotes the predicted predicate label. $S_v$, $S_t$, $S_c$, and $S_o$ denote the visual semantics, structural semantics, conceptual semantics and other semantics, respectively. Specifically, $S_c = S_v \cap S_t$, and $S_o = C_S(S_v \cup S_t)$, which means the complementary set of subset $S_v \cup S_t$ in $S$. $S$ is the whole generalized semantic space. Because the boundary of $S$ is hard to be defined, $S_o$ cannot be explicitly modeled. To solve this problem, $S_o$ is regarded as the global modeling of the generalized semantic space and does not refer to any specific semantics, which is shown as the gray branch in Fig.2. Thus, Eq.(1) can be rewritten as follows:

$$P(SG|I) = P(R|\boldsymbol{r}_{ij}, \boldsymbol{P}_{vis}) + P(R|\boldsymbol{r}_{ij}, \boldsymbol{P}_{str}) + P(R|\boldsymbol{r}_{ij}, \boldsymbol{P}_{con}) + P(R|\boldsymbol{r}_{ij}), \quad (2)$$

where $\boldsymbol{r}_{ij}$ is the relation representation. $\boldsymbol{P}_{vis}$, $\boldsymbol{P}_{str}$, and $\boldsymbol{P}_{con}$ denote the visual prototype, structural prototype and conceptual prototype, respectively. The pipeline of our method is shown in Fig.2. For the $i$-th proposal, the appearance feature $\boldsymbol{a}_i$ is the region visual feature extracted by the object detector, and the structural semantic embedding of the entity label is denoted as $\boldsymbol{s}_i$, which is the word embedding from Glove [32]. The visual feature of the closed union bounding box is denoted as $\boldsymbol{v}_{ij}$, and the original relation feature is denoted as $\boldsymbol{r}_{ij}$.

## 3.2 Collaboration-induced Prototype Learning

To extract the robust interaction between visual and structural semantic subspaces, the structural prototype and the visual prototype are modeled individually. Then a Residual-Driven Representation Enhancement (RDRE) module is designed to capture the visual details from the prototype.

*3.2.1 Relation Feature Extraction.* As Fig.2 shown, the relation feature is extracted based on the detected results of object detector. In many typical SGG methods [7, 36, 44], this module is composed of an object encoder for entity-level context extraction, an object decoder for entity label prediction, a relation encoder for relation-level context extraction and a relation decoder for predicate label prediction. The common object and relation encoders include RNN, GNN, and Transformer, which are with high computational complexity.

However, in recent works, this complex framework is greatly simplified without any context extraction, and the simple operations, such as concatenation and element-wise product, are adopted to fuse the visual semantics and structural semantics. Here we further simplify this extraction module and only $\boldsymbol{s}_i$ and $\boldsymbol{a}_i$ are utilized to get the relation feature $\boldsymbol{r}_{ij}$. The whole process is defined as follows:

$$\boldsymbol{g}_i = \sigma(\boldsymbol{W}_g([\boldsymbol{W}_s\boldsymbol{s}_i, \boldsymbol{W}_a\boldsymbol{a}_i])), \quad (3)$$

$$\tilde{\boldsymbol{r}}_i = (\boldsymbol{s}_i + \boldsymbol{s}_i \odot \boldsymbol{g}_i), \quad (4)$$

$$\boldsymbol{r}_{ij} = F(\tilde{\boldsymbol{r}}_i^s, \tilde{\boldsymbol{r}}_j^o), \quad (5)$$

where $\boldsymbol{W}_g$, $\boldsymbol{W}_s$, $\boldsymbol{W}_a$ are learnable matrices, $\sigma$ denotes the sigmoid function, and $[\cdot]$ is the concatenation operation. $\odot$ is the element-wise product. $\tilde{\boldsymbol{r}}_i^s$ and $\tilde{\boldsymbol{r}}_i^o$ denote the subject feature and object feature according to the candidate pairs. $F(\cdot)$ is fusion function proposed by [47].

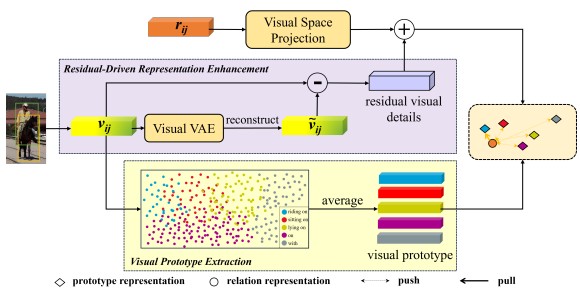

**Figure 3: The illustration of the proposed Residual-Driven Representation Enhancement (RDRE) module. The difference between $v_{ij}$ and $\tilde{v}_{ij}$ is modeled as residual, which is encoded and added to the visual space representation.**

*3.2.2 Visual Prototype Modeling.* To represent the visual semantics comprehensively, the visual prototype is required to contain the following information, i.e. the appearance of subject and object, and the spatial relation between them. Spatial feature plays a critical role in visual semantics. For example, the predicate "riding" contains positional cues naturally. That is, for the triplet <subject-*riding*-object>, it indicates the subject is above and the object is below. And it is nearly impossible to find a situation where the subject and the object are next to each other. A practice in previous works [7, 36, 44] to model this spatial feature is to encode the proposed bounding box information, including coordinates, area, etc, into context, which is integrated with union visual feature $v_{ij}$ to obtain visual semantics. In fact, $v_{ij}$ itself can satisfy the two requirements that visual prototype raised, and the spatial cues are implicitly modeled as follows:

$$a_{ij} = FM(union(b_i, b_j)), \tag{6}$$

$$t_{ij} = Conv([rec_i, rec_j]), \tag{7}$$

$$v_{ij} = MLP(a_{ij} + t_{ij}), \tag{8}$$

where $FM(\cdot)$ is the feature map output by the object detector, $i$ and $j$ denote the proposal indexes of subject and object, $union(\cdot)$ is the function to compute the union region of two bounding boxes $b_i$ and $b_j$, and $a_{ij}$ is the appearance feature of union region from feature map. $Conv(\cdot)$ is the convolution layer. $rec_i$ is the mask of corresponding region on the feature map, and for the internal pixel of bounding box, the value is set to 1, otherwise, it is set to 0. Thus, the spatial feature is introduced in $t_{ij}$. After a simple integration with appearance feature $a_{ij}$, the union visual feature $v_{ij}$ is obtained.

To extract visual prototypes, $v_{ij}$ of all foreground paired entities in training samples are recorded and summed according to annotated predicate categories. Meanwhile, the sample numbers of each predicate category are recorded. The visual prototype of each foreground predicate is averaged by the corresponding sample number, which is shown in the yellow part of Fig.3. Because background triplets without annotations are ignored in this process, the visual prototype of the predicate "background" is obtained by the Gaussian distribution $N(0, 1)$.

To emphasize the equal importance of visual and structural semantics, the prototypes in these two subspaces are modeled independently. The structural semantic prototype is extracted by the

average word embedding of all words in a predicate. The relation feature $r_{ij}$ is projected into different semantic subspace by MLP. The projected representation is denoted as $r_{str}$ and $r_{vis}$ respectively, of which the similarity with the corresponding prototype is calculated.

*3.2.3 Residual-Driven Representation Enhancement.* There is another problem for $r_{vis}$ to match with the corresponding visual prototype. Specifically, the visual prototype is based on the union visual feature $v_{ij}$, but in our relation feature extraction module, there is not union information. $v_{ij}$ contains a lot of useful visual details in the background, which should be captured. To introduce these details into $r_{vis}$, we propose this Residual-Driven Representation Enhancement (RDRE) module, leveraging the powerful modeling ability of VAE [17] in latent space. Different from the common use of VAE such as generating new samples and classifying, our RDRE utilizes it to model the details. As Fig.3 shown, the visual VAE in RDRE is modeled to reconstruct the input feature $v_{ij}$ as much as possible, but it cannot be exactly the same as the output feature $\tilde{v}_{ij}$. This is because the latent space is low dimensional and the compression makes some detailed information lost. The difference between $v_{ij}$ and $\tilde{v}_{ij}$ is regarded as the residual, which should be focused on by $r_{vis}$. The process of RDRE supplementing details contained in the visual prototype to the visual representation is as follows:

$$\tilde{v}_{ij} = VAE_v(v_{ij}), \tag{9}$$

$$\tilde{r}_{vis} = r_{vis} + MLP(v_{ij} - \tilde{v}_{ij}). \tag{10}$$

The VAE loss of visual VAE is composed of reconstruction loss and distribution alignment loss, which is defined as follows:

$$L_{VAE\_v} = ||r_{vis} - \tilde{r}_{vis}||_2 + D_{KL}(N(\mu_v, \sigma_v^2), N(\mu_s, \sigma_s^2)), \tag{11}$$

where $|| \cdot ||_2$ is the function to calculate L2-norm. $D_{KL}(\cdot)$ is to calculate the Kullback-Leibler (KL) divergence, $\mu_v$ and $\sigma_v$ are the mean and variance of the latent space of visual VAE, and $\mu_s$ and $\sigma_s$ are the mean and variance of standard normal distribution.

## 3.3 Intersection-induced Prototype Learning

After the interaction between visual and structural semantics is explored, the intersection between them is modeled as conceptual semantics with the conceptual prototype, which is enhanced by a new conceptual VAE.

*3.3.1 Conceptual Prototype Modeling.* For a predicate category, although the representations of visual and structural semantics are different, they all point to the same concept, which may be modeled as conceptual semantics to represent this intersection. It is a higher-level semantics, which enables the method to generalize from specific instances to broader principles. The definitions in the dictionary are sufficiently credible as the concept, because they are based on consensus and can reflect how words are commonly used and understood by speakers of the language. For example, for the predicate "between", the definition in Wiktionary is *In the position or interval that separates **two things***. If the meaning of "two things" is understood accurately by the model, many confusing candidate predicates regarding "between" can be identified.

Due to several long sentences of definitions in a concept, how to model these flexible and ultra-long texts is a critical problem

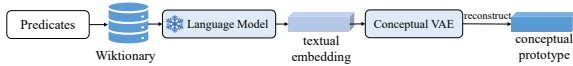

Figure 4: The illustration of Intersection-induced Prototype Learning (IPL) method. The original conceptual prototype of each predicate is extracted by a frozen LLM, which is reconstructed by a conceptual VAE to discard the irrelevant detailed information.

---

**Algorithm 1** Algorithm of Definition Filtration.

---

**INPUT:** All predicate labels $R$ in the dataset.
**OUTPUT:** Filtered Definitions $D$ of predicate labels.
**for** each label $r \in R$: **do**
    Set all the candidate definitions as $d_a$, $d_a \leftarrow \phi$.
    Set the filtered definitions as $D_r$, $D_r \leftarrow \phi$.
    **if** $r$ is verb:
        Remove the tense of $r$.
    **end if**
    **if** $r$ in Wiktionary:
        Add corresponding definitions to $d_a$.
    **else**:
        **for** each word $w_j \in r$: **do**
            Set the definitions of $w_j$ as $d_p$; $d_a \leftarrow \{d_a, d_p\}$
        **end for**
    **end if**
    **for** each definition $def$ in $d_a$: **do**
        **if** $def$ is for verb or preposition or conjunction:
            $D_r \leftarrow \{D_r, def\}$.
        **end if**
    **end for**
    Reserve top $k$ definitions in $D_r$.
**end for**
**if** $r_m$ and $r_n$ are with different tenses:
    Replace the last definition in $D_m$ and $D_n$ with an example sentence corresponding to $r_m$ and $r_n$, respectively.
**end if**
**return** $D = \{D_1, D_2, ..., D_r, ...\}$

---

to be solved. Thanks to the rapid development of LLM, the input sequences are not limited to fixed length and the long-range dependency is better solved compared with the conventional Transformer. However, there are several definitions of a concept, covering various part of speech, such as verb, noun, conjunction, etc. To preserve the most meaningful part and make the semantically similar predicates distinguishable, the definitions are filtered and supplemented as Algorithm 1 shown.

*3.3.2 Essence-Driven Prototype Enhancement.* Although the most meaningful definitions are preserved by Algorithm 1, there are still many irrelevant words in definitions, such as pronouns, to be filtered. To make the method concentrate more on the essence of each concept, a conceptual VAE is utilized to reconstruct the textual embedding from LLM. The information that cannot be reconstructed is also regarded as details. But different from the visual VAE, these details are discarded, because the conceptual prototype needs to be

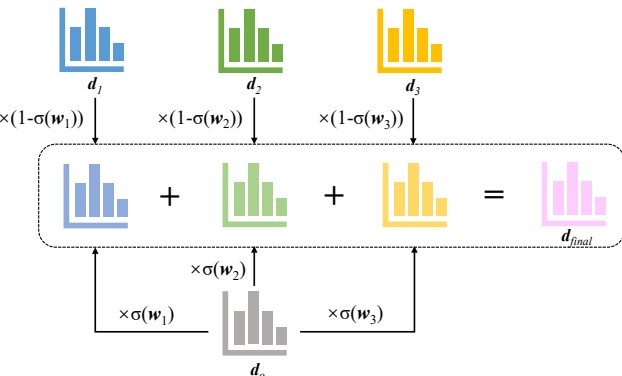

Figure 5: The illustration of Selective Fusion Module (SFM). The output of three branches, i.e. conceptual semantics (blue), structural semantics (green), visual semantics (yellow), and generalized semantics projection (gray), are integrated according to the decomposed generalized semantic space.

compact to provide a clear guide for the projected representation in conceptual semantic space. The final conceptual prototype is the reconstructed feature by the conceptual VAE. The corresponding VAE loss is as follows:

$$L_{VAE\_c} = ||\boldsymbol{P}_{con} - \tilde{\boldsymbol{P}}_{con}||_2 + D_{KL}(N(\mu_c, \sigma_c^2), N(\mu_s, , \sigma_s^2)), \quad (12)$$

where $\boldsymbol{P}_{con}$ is the original textual embedding extracted by LLM, and $\tilde{\boldsymbol{P}}_{con}$ is reconstructed by the conceptual VAE. $\mu_c$ and $\sigma_c$ are the mean and variance of the latent space of conceptual VAE.

## 3.4 Selective Fusion Module

As shown in Eq.(2), other semantics $S_o$ is modeled as the generalized semantics projection, which is a global modeling of the generalized semantic space. The output distribution of this branch $\boldsymbol{d}_o$ is calculated by relation feature $\boldsymbol{r}_{ij}$ fed into the classifier. The cosine similarities between the prototype and representation in the visual, structural, and conceptual semantic subspaces are calculated as the following:

$$\boldsymbol{d}_1 = \frac{<\tilde{\boldsymbol{r}}_{vis}, \boldsymbol{P}_{vis}>}{||\tilde{\boldsymbol{r}}_{vis}||_2||\boldsymbol{P}_{vis}||_2}, \quad (13)$$

$$\boldsymbol{d}_2 = \frac{<\boldsymbol{r}_{str}, \boldsymbol{P}_{str}>}{||\boldsymbol{r}_{str}||_2||\boldsymbol{P}_{str}||_2}, \quad (14)$$

$$\boldsymbol{d}_3 = \frac{<\boldsymbol{r}_{con}, \tilde{\boldsymbol{P}}_{con}>}{||\boldsymbol{r}_{con}||_2||\tilde{\boldsymbol{P}}_{con}||_2}, \quad (15)$$

where $< \cdot >$ means the dot product of two vectors.

The above three similarities are regarded as the output distributions from different semantic subspaces. Then the results of generalized semantics projection branch and three semantic subspace-based branches are integrated as Fig.5 shown. Three learnable factors $\boldsymbol{w}_1$, $\boldsymbol{w}_2$, and $\boldsymbol{w}_3$ are utilized to dynamically measure the importance of the semantic subspace based result and the global result. The final distribution $\boldsymbol{d}_{final}$ for predicate classification is as follows:

$$\boldsymbol{d}_{final} = \sum_{k=1}^{3} \boldsymbol{d}_o \times \sigma(\boldsymbol{w}_k) + \boldsymbol{d}_k \times (1 - \sigma(\boldsymbol{w}_k)). \quad (16)$$

## 3.5 Loss Fuction

To make the prototype of different predicate categories distinguish from each other, a simple but effective similarity loss $L_{sim}$ is proposed. Take the conceptual prototype $\tilde{P}_{con}$ as an example, the similarity matrix is calculated as follows:

$$S = \tilde{P}_{con} \times \tilde{P}_{con}^T, \tag{17}$$

where $\tilde{P}_{con} \in N \times D$, $N$ is the number of predicate categories, and $D$ is the dimension of prototype. $S$ is the similarity matrix, which is a square matrix with all diagonal elements are 1. The other elements in $S$ mean the similarity between different pairwise prototypes, which should be minimized. This similarity loss is defined as follows:

$$L_{sim} = [sum(S) - tr(S)]/(N \times N), \tag{18}$$

where $sum(\cdot)$ is the function that sums all elements in a matrix, and $tr(\cdot)$ is to calculate the trace of the matrix. The final loss function $L$ for SPLN is expressed as follows:

$$L = \sum_{k=1}^{3} L_{sim}^k + L_{VAE\_v} + L_{VAE\_c} + L_{obj} + L_{rel}, \tag{19}$$

where $L_{sim}^k$ is the similarity loss of the $k$-th semantic subspace to prevent holistic supervision signal. $L_{obj}$ and $L_{rel}$ are the cross-entropy loss for object and predicate classification respectively.

## 4 EXPERIMENTS

### 4.1 Experimental Settings

We conducted experiments on two datasets, i.e. Visual Genome (VG) [18] and GQA [11]. VG is the most popular dataset in SGG, which contains 150 object classes and 50 predicate classes. GQA is a cross-modal dataset with 200 object classes and 100 predicate classes. All experiments are evaluated on three standard modes: Predicate Classification (PredCls), Scene Graph Classification (SG-Cls), and Scene Graph Detection (SGDet). For the evaluation metric, mean Recall@K (mR@K) is chosen, which is designed specifically for unbiased SGG as the average value of Recall@K (R@K) of all predicate categories.

### 4.2 Implementation Details

Following previous works [20, 39, 45], a pre-trained Faster R-CNN [34] with the ResNeXt-101-FPN backbone is utilized as the object detector, of which parameters are frozen during training. The network is optimized by an SGD optimizer with an initial learning rate of 5e-4. The batch size is set to 8 as many current methods, and each model is trained with 45,000 iterations. The preserved definition number $k$ in Algorithm 1 is set to 3 by default. As previous works [47], to make a trade-off between head and tail predicates, an off-the-shelf re-weighting method [6] is equipped. Here two pre-trained language models, CLIP [33] and LLaMA [37], are adopted to extract the textual embedding of each concept. For CLIP, only the text encoder is utilized. All experiments are conducted on an NVIDIA GeForce RTX 4090 GPU.

### 4.3 Comparisons with State-of-the-art Methods

**Visual Genome.** As Table.1 shown, our SPLN is divided into two versions: SPLN(C) and SPLN(L), where C means the text encoder of CLIP is adopted to extract the textual embeddings of concepts, and L means LLaMA is adopted. Compared with CLIP, LLaMA-based SPLN can achieve better performance, which demonstrates that better conceptual prototypes contribute to more fine-grained scene graphs. All the methods we choose to be compared are the latest proposed, which can be divided into model-agnostic methods and specific methods. Following previous works [14, 16, 24, 42], the model-agnostic methods under two commonly used baseline networks, MOTIFS [44] and VCTree [36], are compared. Our SPLN is a specific method, which achieves the best performance for PredCls. For SGCls, SPLN(L) is the best on mR@20/100, and is only lower than VCTree-EICR [31] by 0.1% on mR@50. For SGDet, SPLN(L) achieves the second highest mR@50/100, and CI-SGG+SI-CGG [3] achieves the highest. It is because CI-SGG+SI-CGG introduces external commonsense knowledge, which provides strong and reasonable priors. However, it leads to extra bias in commonsense space and affects the performance for PredCls and SGCls.

**GQA.** As Table 2 shown, our SPLN not only achieves SOTA performance on VG dataset but also on GQA dataset, which demonstrates the generalization of our method. Our SPLN(C) outperforms other methods for the most challenging subtask SGDet, leading to a new state-of-the-art performance. Inf [2] performs better than our SPLN(L) for PredCls on mR@100, but it is sensitive to dataset changes. For the experiments conducted on VG dataset, the mR@100 for PredCls of VCTree-Inf and MOTIFS-Inf is 30.7%, while our SPLN(C) is 47.1%, and SPLN(L) is 48.5%.

### 4.4 Ablation Studies

**Ablation on Different Prototypes.** As Table.3 shown, the effectiveness of each prototype in the corresponding semantic subspace is demonstrated. All the experiments are conducted on the VG dataset for PredCls. The result of baseline is from the single generalized semantics projection branch, and the other ablation results are obtained with this branch integrated. Because our method is equipped with the reweighting strategy [6], the baseline method is equipped with the same strategy for a fair comparison.

When a single type of prototype is utilized, the performance is limited. Especially when only visual prototype is utilized, performance is damaged compared to the baseline. This is because the problem of ambiguous decision boundaries is caused by diverse visual features, and the visual prototype suffers from the same problem and makes it worse. When working together with the structural semantic prototype or the conceptual prototype, the performance is improved significantly, indicating the ambiguity is relieved. When all branches are utilized simultaneously, the generalized semantic space is modeled comprehensively, and performance achieves the best, which validates the effectiveness of the interaction and intersection among the decomposed semantic subspaces.

**Ablation on Model Components.** As Table.4 shown, the performances of each component are evaluated, which can be observed that every component in SPLN is indispensable. Results show the importance of RDRE and EDPE, which demonstrates the necessity of representation and prototype reconstruction. If the visual representation and conceptual prototype are not enhanced, the performance will drop. Meanwhile, the ablation of SFM is conducted.

Table 1: Performance (%) comparison of our method and the state-of-the-art methods on VG dataset. The best result is marked in bold and the second best result is marked with underline. † means using the resampling strategy provided by [45], and ‡ means using the resampling strategy provided by [25].

| Type | Models | Source | Predicate Classification mR@20 | mR@50 | mR@100 | Scene Graph Classification mR@20 | mR@50 | mR@100 | Scene Graph Detection mR@20 | mR@50 | mR@100 |
|---|---|---|---|---|---|---|---|---|---|---|---|
| Model-Agnostic / MOTIFS | baseline [44] | CVPR18 | 10.8 | 14.0 | 15.3 | 6.3 | 7.7 | 8.2 | 4.2 | 5.7 | 6.6 |
| | A-PFG [39] | AAAI23 | 33.5 | 39.9 | 42.0 | 19.8 | 22.7 | 23.6 | 11.9 | 15.8 | 18.3 |
| | BAI [23] | ACM MM23 | 28.7 | 34.3 | 36.5 | 14.3 | 16.2 | 17.2 | 10.2 | 15.2 | 17.1 |
| | SIL [38] | ACM MM23 | 13.6 | 16.9 | 18.4 | 8.5 | 10.2 | 10.8 | 5.5 | 7.3 | 8.5 |
| | DKBL [5] | ACM MM23 | — | 29.7 | 32.2 | — | 18.2 | 19.4 | — | 12.6 | 15.1 |
| | Inf [2] | CVPR23 | — | 24.7 | 30.7 | — | 14.5 | 17.4 | — | 9.4 | 11.7 |
| | EICR [31] | ICCV23 | — | 34.9 | 37.0 | — | 20.8 | 21.8 | — | 15.5 | 18.2 |
| | CFA [20] | ICCV23 | — | 35.7 | 38.2 | — | 17.0 | 18.4 | — | 13.2 | 15.5 |
| | LS-KD [24] | TCSVT23 | — | 34.2 | 37.9 | — | 18.7 | 20.9 | — | 13.7 | 16.6 |
| | QuatRE [42] | TMM23 | 17.7 | 23.9 | 28.0 | 10.7 | 14.8 | 17.1 | 6.4 | 8.9 | 10.9 |
| | ST-SGG† [16] | ICLR24 | — | 32.5 | 35.1 | — | 18.0 | 19.3 | — | 12.9 | 15.8 |
| VCTree | baseline [36] | CVPR19 | 14.0 | 17.9 | 19.4 | 8.2 | 10.1 | 10.8 | 5.2 | 6.9 | 8.0 |
| | A-PFG [39] | AAAI23 | 34.6 | 41.7 | 43.8 | 17.3 | 20.4 | 21.8 | 12.3 | 16.6 | 19.1 |
| | BAI [23] | ACM MM23 | 28.2 | 32.8 | 34.6 | 16.0 | 18.9 | 19.6 | 9.6 | 12.6 | 16.8 |
| | SIL [38] | ACM MM23 | 14.5 | 18.1 | 19.6 | 10.1 | 12.6 | 13.9 | 5.7 | 7.6 | 8.9 |
| | DKBL [5] | ACM MM23 | — | 28.7 | 31.3 | — | 21.2 | 22.6 | — | 11.8 | 14.2 |
| | Inf [2] | CVPR23 | — | 28.1 | 30.7 | — | 17.3 | 19.4 | — | 10.4 | 11.9 |
| | EICR [31] | ICCV23 | — | 35.6 | 37.9 | — | **26.2** | 27.4 | — | 15.2 | 17.5 |
| | CFA [20] | ICCV23 | — | 34.5 | 37.2 | — | 19.1 | 20.8 | — | 13.1 | 15.5 |
| | LS-KD [24] | TCSVT23 | — | 33.8 | 37.5 | — | 22.2 | 24.9 | — | 13.7 | 16.6 |
| | QuatRE [42] | TMM23 | 19.3 | 25.5 | 29.1 | 10.8 | 15.9 | 18.7 | 7.0 | 9.5 | 11.6 |
| | ST-SGG† [16] | ICLR24 | — | 32.7 | 35.6 | — | 21.0 | 22.4 | — | 12.6 | 15.1 |
| Specific | Het-SGG [43] | AAAI23 | — | 31.6 | 33.5 | — | 17.2 | 18.7 | — | 12.2 | 14.4 |
| | CI-SGG+SI-CGG [3] | ACM MM23 | — | 42.8 | 45.0 | — | 25.4 | 27.1 | — | **19.3** | **21.5** |
| | PE-Net [47] | CVPR23 | — | 31.5 | 33.8 | — | 17.8 | 18.9 | — | 12.4 | 14.5 |
| | CV-SGG [14] | CVPR23 | — | 32.6 | 36.2 | — | — | — | — | 14.6 | 17.0 |
| | SQUAT‡ [15] | CVPR23 | 25.6 | 30.9 | 33.4 | 14.4 | 17.5 | 18.8 | 10.6 | 14.1 | 16.5 |
| | IS-GGT [19] | CVPR23 | — | 26.4 | 31.9 | — | 15.8 | 18.9 | — | 9.1 | 11.3 |
| | TGIR [40] | IJCV23 | 12.7 | 16.1 | 17.2 | 7.6 | 9.2 | 9.7 | 4.4 | 6.4 | 7.9 |
| | CSL [28] | TPAMI23 | — | 29.5 | 31.6 | — | 16.7 | 17.9 | — | 11.9 | 14.3 |
| | SPLN(C) | — | 36.4 | 43.9 | 47.1 | 21.3 | 25.6 | 27.4 | 12.4 | 17.2 | 20.6 |
| | SPLN(L) | — | **36.9** | **44.9** | **48.5** | **21.9** | 26.1 | **28.0** | **13.6** | 17.6 | 20.8 |

The experiment of w/o SFM means the output distributions from three semantic branches are simply summed without generalized semantics projection involved. This verifies the effectiveness of regarding generalized semantics projection as other semantics to model the generalized semantic space. The importance of $L_{sim}$ is also evaluated, which helps our SPLN to make more distinguishable prototypes.

## 4.5 Qualitative Analysis

***Visualization of R@K***. As Fig.6 shown, Recall@100s of all predicate categories are reported, which are sorted in descending order by frequency. The experiments are conducted on the VG dataset under PredCls task, and the baseline is equipped with the same re-weighting strategy as our SPLN. As shown in many unbiased works [15, 16, 45], it is a common phenomenon that the performance of head predicates decreases. This is because the trivial head predicates are divided into fine-grained body and tail predicates with more semantics. Our SPLN(L) maintains the head performance as much as possible, and the drop is limited within a small margin. Meanwhile, our SPLN(L) performs well for the body and tail predicates, which effectively solves the long-tailed problem. For the predicate "flying in", which only has four samples during training, the R@K in the testing set is improved from 0 to 64.91% by our method.

***Visualization of Image***. As Fig.7 shown, the visualization of the generated scene graphs of baseline and our SPLN(L) is compared.

**Table 2: Performance comparison of our method and the state-of-art-methods on GQA dataset(%). The best result is marked in bold and the second best result is marked with underline.**

| Type | Models | PredCls mR@50/100 | SGCls mR@50/100 | SGDet mR@50/100 |
|---|---|---|---|---|
| Model-Agnostic / MOTIFS | baseline [44] | 16.4 / 17.1 | 8.2 / 8.6 | 6.4 / 7.7 |
| | Inf [2] | 37.9 / 40.1 | 19.1 / 20.0 | 14.3 / 15.8 |
| | EICR [31] | 36.3 / 38.0 | 17.2 / 18.2 | 16.0 / 18.0 |
| | CFA [20] | 31.7 / 33.8 | 14.2 / 15.2 | 11.6 / 13.2 |
| Model-Agnostic / VCTree | baseline [36] | 16.6 / 17.4 | 7.9 / 8.3 | 6.5 / 7.4 |
| | Inf [2] | **39.4** / **41.6** | 19.2 / 20.0 | 13.6 / 15.1 |
| | EICR [31] | 35.9 / 37.4 | 17.8 / 18.6 | 14.7 / 16.3 |
| | CFA [20] | 33.4 / 35.1 | 14.1 / 15.0 | 10.8 / 12.6 |
| Specific | SPLN(C) | 38.2 / 39.6 | 17.8 / 18.4 | 16.2 / **18.6** |
| | SPLN(L) | **39.4** / 40.7 | **19.4** / **20.2** | **16.4** / 18.4 |

**Table 3: Ablation studies of each prototype of SPLN(C) (%).**

| Structural | Visual | Conceptual | mR@20 | mR@50 | mR@100 |
|---|---|---|---|---|---|
| | baseline | | 32.7 | 40.1 | 43.7 |
| ✓ | | | 34.2 | 41.8 | 45.1 |
| | ✓ | | 32.2 | 39.3 | 42.3 |
| | | ✓ | 33.3 | 40.4 | 43.8 |
| ✓ | ✓ | | 35.5 | 43.1 | 46.4 |
| ✓ | | ✓ | 34.9 | 42.2 | 45.5 |
| | ✓ | ✓ | 35.1 | 42.6 | 46.3 |
| ✓ | ✓ | ✓ | **36.4** | **43.9** | **47.1** |

**Table 4: Ablation studies of components of SPLN(C) (%).**

| Models | mR@20 | mR@50 | mR@100 |
|---|---|---|---|
| w/o RDRE | 35.7 | 42.9 | 45.8 |
| w/o EDPE | 35.3 | 42.7 | 45.9 |
| w/o SFM | 35.5 | 43.3 | 46.5 |
| w/o $L_{sim}$ | 36.0 | 42.7 | 46.0 |
| SPLN(C) | **36.4** | **43.9** | **47.1** |

The red arrows denote the wrong or coarse-grained predictions, and the green arrows denote the right or fine-grained predictions. The baseline is strong with high mR@K values, but the visualization shows that it overfits the body and tail predicates, leading to some unreasonable predictions such as "train-to-window" in the first row and "arm1-part of-fence" in the second row. Compared to the baseline, our SPLN(L) shows a more reasonable trade-off between head and tail predicates, which does not deliberately suppress the prediction of head predicates such as "train-has-window" in the first row. Furthermore, our method generates more semantically rich triplets such as "window-part of-building" v.s. "window-on-building" and "man-riding-skateboard" v.s. "man-using-skateboard" in the second row.

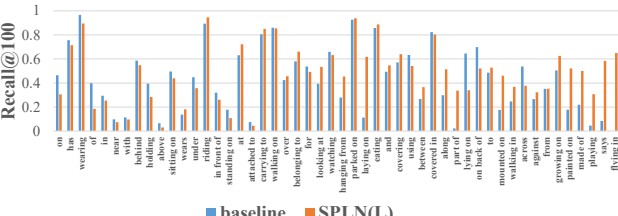

**Figure 6: Recall@100 of baseline and our SPLN(L) for all predicate categories under PredCls task on the VG dataset.**

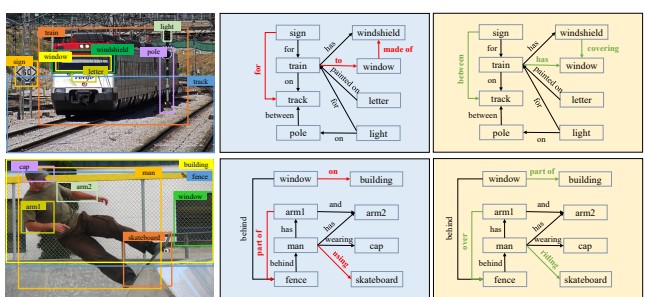

**Figure 7: Visualization of the generated scene graphs by baseline (in blue) and our SPLN(L) (in yellow) under PredCls task on the VG dataset.**

## 5 CONCLUSION

In this work, a Synergetic Prototype Learning Network (SPLN) is proposed to deal with the ambiguous decision boundaries by modeling the generalized semantic space, which is decomposed as visual semantics, structural semantics, conceptual semantics, and other semantics. The interaction between visual and structural semantics is comprehensively explored. To bridge the gap between the visual prototype and corresponding representation, a Residual-Driven Representation Enhancement module is proposed, which regards the residual that cannot be modeled by VAE as details and models them explicitly. The intersection between visual and structural semantics is modeled the conceptual semantics, which adopts the definitions in a Wiktionary. To preserve the essential information in the conceptual prototype, an Essence-Driven Prototype Enhancement module is proposed, which discards the irrelevant details. Due to the difficulty to define the boundary of generalized semantic space, other semantics is modeled by generalized semantics projection, which is a global modeling and does not refer to any specific semantics. Finally, to integrate the output distributions of these three branches and the generalized semantics projection, a Selective Fusion Module is proposed to dynamically measure the importance of them, where a simple but effective similarity-based loss is proposed to make the prototypes of predicate categories apart from each other. Experimental results show our SPLN achieves state-of-the-art performance on unbiased metrics. In the future, we try to generate more meaningful scene graphs by exploring more types of synergetic effects among different semantic subspaces.

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
