# OpenReview forum: "Synergetic Prototype Learning Network for Unbiased Scene Graph Generation"
_acmmm.org/ACMMM/2024/Conference — MM2024 Poster_

### Official Review · Reviewer_zH6y · 2024-05-04

**Rating:** 6
**Confidence:** 4

**Summary:**

The paper introduces a Synergetic Prototype Learning Network (SPLN) for unbiased Scene Graph Generation. It addresses the challenge of ambiguous decision boundaries by modeling a generalized semantic space and exploring the interaction between visual and structural semantics. The SPLN incorporates modules for representation enhancement, prototype refinement, and selective fusion to improve scene graph generation performance.

**Strengths:**

Innovative Approach: The SPLN introduces a novel method for addressing ambiguous decision boundaries in Scene Graph Generation by modeling a generalized semantic space.
Comprehensive Exploration: The paper comprehensively explores the interaction between visual and structural semantics, conceptual semantics, and other semantic subspaces to improve scene graph generation accuracy.
State-of-the-Art Performance: Experimental results demonstrate that the SPLN achieves state-of-the-art performance on unbiased metrics, showcasing the effectiveness of the proposed approach.

**Limitations:**

The related work section of this article is somewhat insufficient. The authors need to discuss the latest work in scene graph generation in Chapter 2, including but not limited to:
Panoptic Scene Graph Generation with Semantics-Prototype Learning
Multi-Prototype Space Learning for Commonsense-Based Scene Graph Generation
Adaptive Feature Learning for Unbiased Scene Graph Generation
There is a lack of discussion on future work. It would be beneficial to explore the possibility of using retrieval of external knowledge or pre-trained large models to extend multi-label prediction or open-set prediction in scene graph generation, rather than merely overfitting to a single dataset. The authors could discuss more future work in the paper.

**Suitability:**

3

---

### Official Review · Reviewer_2Mtm · 2024-05-25

**Rating:** 4
**Confidence:** 3

**Summary:**

This paper constructs 3 prototypes for each predicate, including visual prototype, word embedding prototype and a newly proposed concept prototype. It proposes a residual-driven representation enhancement module to preserve details in visual representations, and an essence-driven prototype enhancement module to alleviate unrelated noises in long concept descriptions. It also proposes a selective fusion module to fuse the final results.

**Strengths:**

1. This article discusses the existing issues in SGG and proposes a novel solution based on the predicate concepts by leveraging LLM.
2. The method proposed significantly overcomes the baselines adopted in the paper, evaluated from qualitative and quantitative perspectives.

**Limitations:**

1. This paper introduces several concepts that are not commonly used in SGG or related research papers. However, these concepts are inadequately defined or explained within the paper, presenting significant hindrance to understand. e.g., why ''word embedding of labels'' is defined as ''structural semantics'', but not ''linguistic semantics'' or others.
2. Some of the content is not described clearly. Table 3 indeed demonstrates that the baseline model, which excludes all the core modules proposed in the paper, achieves notable performance compared to previous methods. It is hard to understand why the generalized semantics alone can yield such high performance, since the network design and training is not complex (a Faster R-CNN and an MLP, without additional training data).
3. Headers in Table. 1 is not aligned.

**Suitability:**

2

---

### Official Review · Reviewer_qv9T · 2024-05-27

**Rating:** 4
**Confidence:** 4

**Summary:**

The research of SGG suffered from the issue of the ambiguous decision boundaries caused by diverse appearance features. A Synergetic Prototype Learning Network (SPLN) is proposed in this paper. Main idea is the exploration of Conceptual Semantics. Three branches are designed for the SPLN, that is, CPL,IPL and SFM. Overall, this paper is well-structure and has some novelty. More explanation for the Conceptual Semantics is required.

**Strengths:**

(1) The visual semantics, structural semantics and their intersection are collaborated to find the accurate predicate. It is an interesting approach.
(2) RDRE and EDPE are designed to support the function of CPL and IPL, respectively.
(3) Sufficient experiments are conducted.

**Limitations:**

(1) Although the Conceptual Semantics is introduced in the paper from Line 132, I am still confused with the intersection operation. Generally, visual semantics and structural semantics are derived from visual perception and spatial perception, respectively. How to intersect to form one more abstract conceptual space? Moreover, some concepts have various meaning under different context.
(2) In Figure 2, where is the structural semantics?
(3) From a venn diagram, Conceptual Semantics is obtained from the input of The visual semantics and structural semantics. However, in Figure 2, SRM uses three distinct semantics as input. More introduction is necessary.
(4) Do the authors consider multi-prototype for network learning?

**Suitability:**

3

---

### Meta-Review · Area_Chair_E4WM · 2024-07-01

**Recommendation:** Accept (Poster)
**Confidence:** 4

**Metareview:**

For unbiased scene graph generation, this paper proposes a synergetic prototype learning network. It receives two borderline accept, and one accept. The response well addresses the reviewers' concerns. The merits, including the novel idea, good writing, and sufficient experiments, are well recognized by the reviewers. Please also incorporate the detailed analysis in response into the camera-ready version. Overall, I think this is a good paper and should be accepted.